# Diurnal cycle of short-term fluctuations of integrated water vapour above Switzerland

Klemens Hocke[1], Leonie Bernet[1], Jonas Hagen[1], Axel Murk[1], Matthias Renker[2], and Christian Mätzler[1]

[1]Institute of Applied Physics and Oeschger Centre for Climate Change Research, University of Bern, Bern, Switzerland
[2]Armasuisse, Thun, Switzerland

**Correspondence:** Klemens Hocke (klemens.hocke@iap.unibe.ch)

**Abstract.** The TROpospheric WAter RAdiometer (TROWARA) continuously measures integrated water vapour (IWV) with a time resolution of 6 seconds at Bern in Switzerland. During summer, we often see that IWV has temporal fluctuations during daytime while the night-time data are without fluctuations. The data analysis is focused on the year 2010 where TROWARA has a good data quality without data gaps. We derive the spectrum of the IWV fluctuations in the period range from about 1 to 100 min. The FFT spectrum with a window size of 3 months leads to a serious underestimation of the spectral amplitudes of the fluctuations. Thus, we apply a band pass filtering method to derive the amplitudes as a function of period $T_p$. The amplitudes are proportional to $T_p^{0.5}$. Another method is the calculation of the moving standard deviation with time window lengths from about 1 to 100 min. Here, we get similar results as for the band pass filtering method. At all periods, the IWV fluctuations are strongest during summer while they are smallest during winter. We derive the diurnal variation of the short-term IWV fluctuations by applying a moving standard deviation with a window length of 10 min. The daily cycle is strongest during the summer season with standard deviations up to 0.22 mm at about 14:00 CET. The diurnal cycle disappears during winter time. A similar seasonal behaviour is observed in the diurnal cycle of latent heat flux as provided by the Modern-Era Retrospective Analysis for Research and Applications, version 2 (MERRA-2 reanalysis) at Bern. Further, the 3-day averages of the latent heat flux and the magnitude of the short-term IWV variability show a strong correlation at Bern in 2010 ($r = 0.82$ with 95% confidence interval from 0.75 to 0.87). Thus, we suggest that the diurnal cycle of short-term IWV fluctuations at Bern is mainly caused by large convective heating during daytime in summer.

## 1 Introduction

Atmospheric water vapour is the dominant greenhouse gas and acts like a warm blanket for the Earth. Global warming due to man-made $CO_2$ emissions is amplified by increase of the water vapour concentration in a warmer world. This amplification of global warming due to the so-called water vapour feedback is up to a factor of three (Held and Soden, 2000). The electric dipole of the water molecule is responsible for the large latent heat of vaporization of water and for the strong interaction of

electromagnetic waves with water vapour. Integrated water vapour (IWV) is the main contributor to the wet delay of signals of the Global Navigation Satellite System (GNSS) (Guerova et al., 2016). Further, atmospheric water vapour is the reservoir gas for formation of cloud liquid water and precipitation such as snow, hail and rain which are relevant for weather and climate. The annual variation of integrated water vapour is rather strong at Bern (Switzerland) reaching from about 8 mm (or 8 kg m$^{-2}$)

in winter to 24 mm in summer (Hocke et al., 2017). Ortiz de Galisteo et al. (2014) reported that IWV ranges from 14.5 mm to 20.0 mm in Spain. Long-term monitoring of IWV is essential for detection of regional and global trends of IWV (Morland et al., 2009; Parracho et al., 2018).

The spatio-temporal variability of IWV on scales of less than 10 km and hours was assessed by Steinke et al. (2015) using various instruments and atmospheric numerical simulations. The model runs showed IWV variabilities of the order of 0.4 mm

for differences in space of 3-4 km or time of 10-15 min during the presence of a boundary layer. Passive microwave radiometry can provide a high temporal resolution of 10 seconds for IWV measurements. Steinke et al. (2015) reported about standard deviations of IWV observed by microwave radiometers exceeding 1 mm even at short time scales of a few minutes. To our knowledge there are no other studies on the IWV variability at very short scales from 1 to 20 min. Since the short-term IWV fluctuations are likely connected with atmospheric waves and turbulence, we expect a seasonal and diurnal variation of the

IWV variability which was not investigated by Steinke et al. (2015) and others. However, Steinke et al. (2015) suggested that the short-term IWV variability at time scales of 15 min or less is induced by atmospheric turbulence which was simulated by their high-resolution model.

The TROpospheric WAter RAdiometer (TROWARA) has measured continuously integrated water vapour (IWV) with a time resolution of 6 seconds at Bern in Switzerland since 2009. In the time from 1994 to 2008 the temporal resolution was 10

seconds. Thus, it is no problem to analyse the IWV variability at Bern on time scales from 1 to 100 min as function of local time and season. The short-term IWV variability is possibly connected with the growth of the atmospheric boundary layer during daytime in summer. Convective heating and associated turbulence generate variable vertical winds and circulation cells leading to a variable vertical water vapour flux during daytime. We expect that IWV can significantly change during daytime if the antenna beam of the radiometer transects an updraft or downdraft region in the lower troposphere.

Schlemmer et al. (2011) simulated the atmospheric convection over land in summer at mid-latitudes. They found diurnal cycles in surface net shortwave radiation, latent heat flux, sensible heat flux, convective mass flux, convective available potential energy (CAPE), specific cloud water content, and surface precipitation. These diurnal cycles have maxima around noon or in the afternoon and early evening hours. Numerical simulations (Stull, 1988; Yamada and Mellor, 1975) show that the turbulent kinetic energy (TKE) has a strong diurnal cycle with increases of TKE from 0.05 J kg$^{-1}$ at night-time to about 1 J kg$^{-1}$ at

daytime. The maximum occurs at a height of about 300 m above the surface. This increase of TKE is associated with the presence of a convective mixed layer during daytime reaching from the surface to 1.5 km or higher. Lüdi and Magun (2002) determined turbulence parameters in the lower troposphere by analyzing the scintillations of a microwave link between a transmitter and a receiver.

The diurnal cycle in IWV over Bern was described by Hocke et al. (2017) using hourly data of the TROWARA radiometer.

The diurnal cycle in IWV goes from about -0.5 mm (relative to the daily mean value) in the morning hours to about +0.5 mm

during the evening hours. This IWV variation is less than 5%, and it can be assumed that the diurnal cycle in IWV has no direct influence on the diurnal cycle of short-term IWV variability. Nevertheless, the diurnal cycle in IWV can be understood as a residual upward flux of tropospheric water vapour during daytime so that the accumulation of IWV achieves a maximum in the evening hours.

Using observations of wind profilers, radiometers and lidars, Collaud Coen et al. (2014) derived the climatology of the atmospheric boundary layer at Payerne which is closely located to Bern in the Swiss plateau. They also compared the observed height of the boundary layer to the modeled height. The regional weather model predicted a boundary layer height of about 1800 m during summer from May to August while the observations indicated a boundary layer height of about 1500 m or less. During winter the boundary layer height decreased to about 500 m where the model gave slightly lower values than the observations. Generally, vertical profiles of specific humidity or relative humidity can be used to determine the height of the atmospheric boundary layer as described by Seidel et al. (2010). The boundary layer is assumed to be moister than the free troposphere so that the vertical gradient of humidity becomes minimal (extreme) at the height of the atmospheric boundary layer. This vertical gradient method of humidity also shows that there is a connection between IWV and the moist boundary layer.

The aim of the present study is to provide mean values of the amplitudes of the short-term IWV fluctuations in the period range from 1 to 100 min. These mean values may guide modeling studies about water vapour convection and circulation cells in the lower troposphere. Further, we derive the dependence of the short-term IWV variability on the season and the local time. Section 2 describes the TROWARA radiometer and the weather station of the University of Bern. Section 3 explains the data analysis to obtain the amplitudes or the moving standard deviation of the IWV variability. Section 4 presents the results on the short-term IWV variability and its relation to the latent heat flux at Bern. Concluding remarks are given in section 5.

## 2 Measurement instruments and retrieval

### 2.1 TROWARA

Our study is focused on the IWV observations of the TROpospheric WAter RAdiometer (TROWARA). TROWARA is a dual-channel microwave radiometer, and its design and construction were described by Peter and Kämpfer (1992) and Morland (2002). Two ferrite circulator switches at each frequency channel of the radiometer perform the change from the antenna to the noise diodes. The noise diodes serve as hot and cold reference loads. The developed radiometer model considers the measurements of the reflection and transmission coefficients of all radiometer components including the ferrite switches (Morland, 2002). In addition, a tipping curve calibration is performed by using the variable brightness temperature of the clear sky at different elevation angles of the antenna. The instrument is very stable so that the tipping curve calibration is only required 2 or 3 times per year.

TROWARA measures the vertically-integrated water vapour (IWV) which is also known as precipitable water vapour. Further, TROWARA provides the vertically-integrated cloud liquid water (ILW), also known as liquid water path. The instrument is operated inside a temperature-controlled room on the roof of the building for Exakte Wissenschaften (EXWI) of the University

of Bern (46.95°N, 7.44°E, 575 m a.s.l.). The antenna receives the atmospheric radiation inside the room through a microwave transparent window. This indoor operation of TROWARA allows the measurement of IWV even during rainy periods.

The antenna beam of TROWARA has a full width at half power of 4° and is pointing the sky at an zenith angle of 50° towards south-east. At 1 km above surface, the horizontal diameter of the sounding volume of TROWARA is about 170 m. The view direction is constant, so that short-term temporal variations of the brightness temperature, IWV and ILW are well monitored with a time resolution of 6 seconds. TROWARA's IWV measurement has nearly all-weather capability during day and night-time. The ILW measurement cannot be carried out in presence of rain droplets. Thus, the ILW measurement is restricted to cloud droplets (ILW < 0.4 mm). Details of the TROWARA instrument and the retrieval technique are provided by Cossu et al. (2015) and Mätzler and Morland (2009).

In the following, we briefly explain the measurement principle and the retrieval. The microwave channel of TROWARA at 21.4 GHz has a bandwidth of 100 MHz, and the microwave channel at 31.5 GHz has a bandwidth of 200 MHz. The frequency channel at 31.5 GHz is more sensitive to microwaves from atmospheric liquid water, while the frequency channel at 21.4 GHz is more sensitive to microwaves from water vapour since there is a rotational transition line of water vapour centered at 22.235 GHz.

The radiative transfer equation of a non-scattering atmosphere is

$$T_{B,i} = T_c e^{-\tau_i} + T_{mean,i}(1 - e^{-\tau_i}), \tag{1}$$

where $\tau_i$ is the opacity of the $i$-th frequency channel (e.g., 21 GHz) along the line of sight of the radiometer. $T_{B,i}$ is the observed brightness temperature, and $T_c$ is the brightness temperature of the cosmic microwave background. $T_{mean,i}$ denotes the effective mean temperature of the troposphere (Ingold et al., 1998; Mätzler and Morland, 2009).

The equation 1 can be solved for the opacities

$$\tau_i = -ln\left(\frac{T_{B,i} - T_{mean,i}}{T_c - T_{mean,i}}\right) \tag{2}$$

where the TROWARA observations yield the radiances $T_{B,i}$.

In a plane-parallel atmosphere, the opacity is linearly related to IWV and ILW

$$\tau_i = a_i'' + b_i'' IWV + c_i'' ILW, \tag{3}$$

where the coefficients $a''$ and $b''$ partly depend on air pressure. Mätzler and Morland (2009) showed that the coefficients can be statistically derived by means of coincident measurements of radiosondes and fine-tuned at times of periods with a clear atmosphere. The coefficient $c$" indicates the mass absorption coefficient of cloud water. $c$" depends on temperature (and frequency), but not on pressure. It is derived from the physical expression of Rayleigh absorption by clouds (Mätzler and Morland, 2009). After determination of the coefficients, the opacity measurements at 21 and 31 GHz yield the desired parameters IWV and ILW in equation 3.

TROWARA provides a time series of IWV since 1994 with a time resolution of 10 seconds until end of 2009 and 6 seconds afterwards. The IWV time series have been used for trend analysis (Morland et al., 2009; Hocke et al., 2011). Hocke et al.

(2017) analysed diurnal cycles in IWV, ILW and cloud fraction by using all informations of TROWARA which also has an infrared radiometer channel at 9.5 -11.5 $\mu$m. The present study only uses the IWV measurements of the year 2010 when the performance of TROWARA was very good.

## 2.2 MERRA-2 reanalysis data

The Modern-Era Retrospective Analysis for Research and Applications, version 2 (MERRA-2) is an atmospheric reanalysis provided by NASA's Global Modeling and Assimilation Office (GMAO) (Gelaro et al., 2017). The MERRA-2 variable used in the present study is the latent heat flux (or total latent energy flux) at the grid point (47°N, 7.5°E) which is close to Bern. The time resolution of the latent heat flux is one hour. The grid resolution of MERRA-2 is 0.5° in latitude and 0.625° in longitude. Draper et al. (2018) investigated the surface energy fluxes in MERRA-2 in detail and obtained a positive bias of about 5 Wm$^{-2}$
for the latent heat flux of global land annual averages.

## 3  Data analysis

The amplitude spectra of the temporal IWV fluctuations are computed by three different methods. Firstly, the fast Fourier Transform (FFT) spectrum of the summer 2010 is calculated. The arithmetic mean is removed from the time series of IWV. Then, the FFT spectrum is obtained by folding the IWV time series of summer 2010 with a Hamming window and by applying
zero padding at the beginning and end of the time series. The FFT spectrum does not take into account the intermittency of the short wave trains of the IWV fluctuations.

A better method is the band pass filtering of the IWV time series with a digital non-recursive, finite impulse response (FIR) band pass filter performing zero-phase filtering by processing the time series in forward and reverse directions. The number of filter coefficients corresponds to a time window of three times the central period, and a Hamming window of equal length
has been selected for the filter. Thus, the band pass filter has a fast response time to temporal changes in the data series. The variable choice of the filter order permits the analysis of wave trains with a resolution that matches their scale. The bandpass cutoff frequencies are at $f_c = f_p \pm 10\% f_p$, where $f_p$ is the central frequency. More details about the bandpass filtering are given by Studer et al. (2012).

The third method for the estimation of the strength of the IWV fluctuations is a moving standard deviation where the time
window length is subsequently changed from 0.5 min to 90 min.

## 4  Results

The main effect investigated in the present study is that short-term IWV fluctuations occur at daytime in summer (June - August) while they disappear at night. Figure 1a shows a convective cloud system which appears near to Bern in the afternoon of 28 June 2010. We assume that water vapour convection, turbulence and convection cells induce the short-term fluctuations of
IWV. Figure 1b shows the IWV time series for six days in summer 2010 observed by the TROWARA radiometer. It is obvious

that the short-term IWV fluctuations are enhanced during daytime. This main effect can be better shown if the IWV series is band pass filtered at a period of 15 min. Figure 1c depicts the mean amplitude of the IWV fluctuations in the period range from 10 to 20 min. The amplitude of the 15 min-IWV fluctuations is four times greater during daytime than at night.

## 4.1 Spectra of IWV fluctuations

Figure 2 shows the amplitude spectra derived by FFT analysis (blue) and by the band pass filtering method (black) of the IWV series during summer 2010. It is obvious that the FFT method underestimates the amplitude of the IWV fluctuations by 1-2 magnitudes. Further, the exponent of the power law is -2 for the FFT spectrum and -1 for the band pass filter method. Since the FFT method does not take into account the intermittency of the IWV wave trains, the results of the band pass filter method are better than those of the FFT method.

Figure 3 depicts the seasonal variations of the amplitude spectra of the IWV fluctuations in 2010. As expected the summer spectrum (June - August, black line) is the strongest one. The spectra of spring and fall (red and blue) are close together while the winter spectrum (green) is below the other spectra at periods greater than 2 min. At periods > 7 min the four power spectra fulfill the power law $P \sim f^{-1}$ which is indicated by the magenta line.

    Another method to characterize the IWV fluctuations is the moving standard deviation with a variable time window length.

The time window length is a bit similar to the period. The four seasonal spectra of the standard deviation SD are shown as function of the time window length in Fig. 4. Using the SD method, we find a similar power law $P \sim f^{-1}$ as indicated by the magenta line. The SD values in Fig. 4 are about two times larger than the amplitude values in Fig. 3. In the following, we only present results derived by the SD method. The SD values in Fig. 4 are in a good agreement with those of Fig. 8 in Steinke et al. (2015). In the following, we focus on the third method (SD values) since the standard deviation is most common for the

characterization of the variability of a time series.

## 4.2 Seasonal variation of IWV fluctuations and relation to latent heat flux

Figure 5 shows the diurnal variation of SD for a time window length of 10 min and for the different seasons in 2010. It is obvious that the IWV fluctuations are strongest during summer and weakest during winter. During spring, summer and fall there is a maximum of SD during the afternoon hours (around 14:00 -15:00 CET). During winter (green), there is no clear

maximum or minimum of SD. In addition, there is a noise floor in summer since the SD of the IWV fluctuations does not vanish during nighttime. The noise floor is higher during summer than in winter.

    The seasonal variation of the diurnal cycle of short-term IWV fluctuations is possibly related to the annual variation in solar heating of the Earth's surface. Surface heating leads to increased turbulence, convection, and upward water vapour flux during daytime. Figure 6 shows the diurnal cycle of the latent heat flux in winter, spring, summer and fall as derived from MERRA-2

reanalysis data close to Bern in the year 2010. As expected, the diurnal cycle of latent heat flux is strongest in summer. The maximum latent heat flux is around 13:00 CET. That means the observed diurnal cycle of the short-term IWV fluctuations lags the diurnal cycle of latent heat flux by 1-2 hours. A numerical simulation by Schlemmer et al. (2011) showed that the diurnal variation in solar short wave radiation drives the diurnal cycle of latent heat flux which is followed by the diurnal cycles of

CAPE (convective available potential energy) and the convective mass flux. The maxima of the curves in CAPE and convective mass flux occurred in the afternoon and evening hours.

These observations and simulations suggest that the increase of short-term IWV fluctuations during daytime in summer is due to the diurnal cycle of latent heat flux which is a precondition for an increase of strong and variable convection cells in the afternoon. The up- and downdraft regions of the convection cells are passing the antenna beam of the TROWARA radiometer which consequently measures larger or smaller values of IWV in time distances of about 10 min. In addition, a convection cell is itself time variable: The lifetime of a convection cell is roughly between 30 and 60 minutes (Giaiotti et al., 2007).

It remains an open question why the IWV fluctuations do not show a maximum around noon in winter in Fig.5 though there is a clear maximum in latent heat flux in winter around noon (Fig. 6). It could be that other processes such as friction of surface winds and turbulence may play a role in winter for generation of IWV fluctuations so that latent heat flux is not the dominant factor in winter.

Figure 7 compares the 3-day averages of SD of IWV (moving 10 min-window) in the upper panel with 3-day averages of MERRA-2 latent heat flux at Bern in the lower panel for the year 2010. The correlation coefficient $r$ of SD(IWV) and the latent heat flux is equal to 0.82, and the 95% confidence interval ranges from 0.75 to 0.87. Some of the peaks in the latent heat flux coincidently appear in the SD curve of short-term IWV variability in the upper panel, e.g., the double peak around June 2010. Figure 7 suggests that the variability of the latent heat flux is a contributor to the short-term variability of IWV. Figure 8 provides a scatter plot of SD(IWV) and the latent heat flux. The linear regression line is given by the black solid line. It is obvious that a linear dependence exists between both parameters. Because of instrumental noise in the SD(IWV) values the linear regression line intersects the y-axis at -37.0 Wm$^{-2}$.

## 5   Conclusions

During summer, we often see that IWV has temporal fluctuations during daytime while the night-time data have smaller fluctuations. We derive the spectrum of the IWV fluctuations in the period range from about 1 to 100 min. The FFT spectrum with a window size of 3 months leads to a serious underestimation of the spectral amplitudes of the fluctuations. Thus, we apply a band pass filtering method to derive the amplitudes as a function of period $T_p$. The amplitudes are proportional to $T_p^{0.5}$ corresponding to a power law $P \sim f^{-1}$ where $f$ is the frequency. Another method is the calculation of the moving standard deviation (SD) with time window lengths from about 1 to 100 min. Here, we get similar results as for the band pass filtering method. At all periods, the IWV fluctuations are strongest during summer while they are smallest during winter. However, the mean SD value is smaller than 0.5 mm for time window lengths less than 90 min. We derive the diurnal variation of the short-term IWV fluctuations by applying a moving standard deviation with a window length of 10 min. The daily cycle is strongest during the summer season with standard deviations up to 0.22 mm at about 14:00 CET. The diurnal cycle disappears during winter time. A similar seasonal behaviour is obvious in the diurnal cycle of latent heat flux at Bern as provided by MERRA-2 reanalysis. Further, the curves of SD(IWV) and the latent heat flux yield a correlation coefficient $r$ of 0.82 at Bern in 2010 while the 95% confidence interval ranges from 0.75 to 0.87.

Thus, we suggest that the diurnal cycle of the short-term IWV fluctuations is mainly caused by the diurnal cycle of latent heat flux which is a precondition for strong and variable convection cells in the afternoon during summer. The spatio-temporal variability of the convection cells induces the diurnal cycle of short-term IWV fluctuations as observed by the TROWARA radiometer at Bern in summer. However, other sources such as eddies in the lower troposphere may also contribute to the short-term variablility of IWV. High resolution modelling of the diurnal cycle of short-term IWV fluctuations could be compared to the TROWARA observations so that one can estimate how good convective processes are represented by the model. Generally, we think that the high resolution IWV observations of TROWARA can contribute to research on the atmospheric boundary layer.

*Code availability.* Programs are available from KH upon request.

*Data availability.* High resolution IWV data of TROWARA are available upon request. Data of the EXWI weather station are provided by the startwave H2O database (http://www.iapmw.unibe.ch/research/projects/STARTWAVE/).

*Author contributions.* KH performed the data analysis. All authors contribute to the interpretation of the results.

*Competing interests.* We have no competing interests.

*Acknowledgements.* We thank the reviewers and editor for their efforts. The study is supported by SNSF 200021-165516.

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

a)

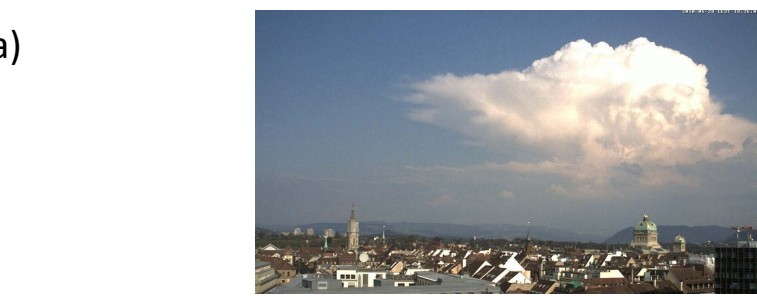

b)

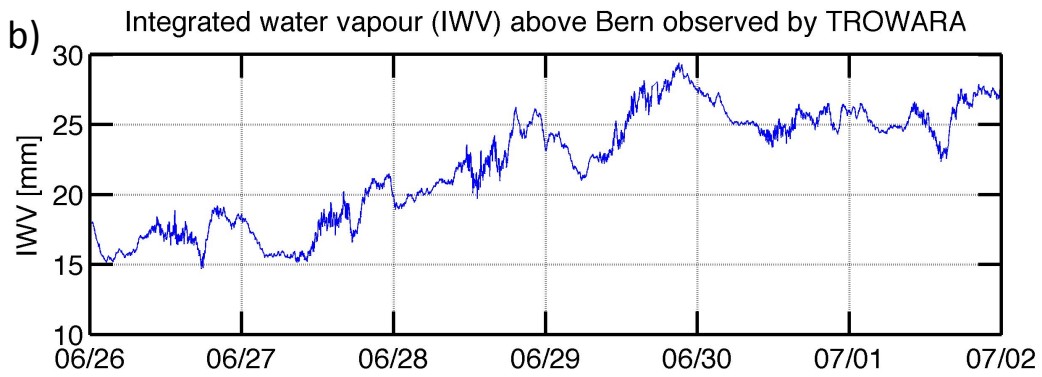

c)

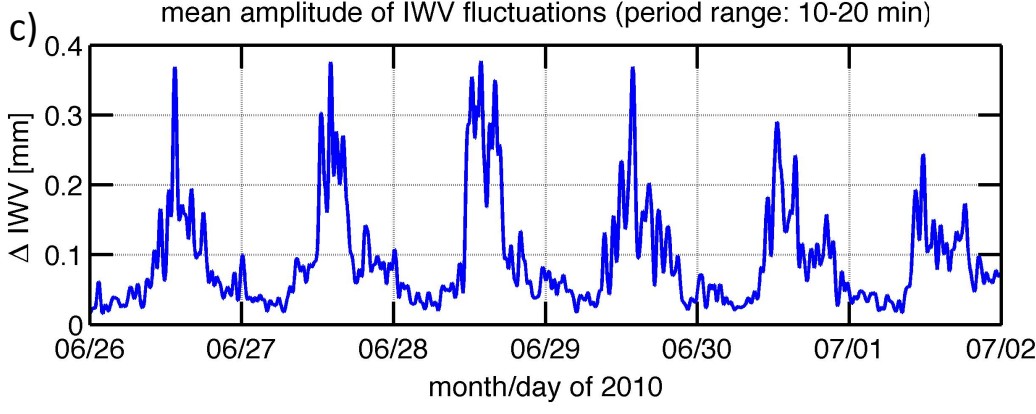

**Figure 1.** a) The convection cell built up within 45 minutes in the south of Bern on June, 28 2010 (picture was taken at 16:36 UT). b) Integrated water vapour (IWV) as function of time (month/day) observed by the Tropospheric Water Radiometer (TROWARA) at Bern. IWV fluctuations regularly occurred between 10 and 18 UT. c) Mean amplitudes of the bandpass-filtered IWV fluctuations (period range 10 to 20 min) show a strong diurnal variation. Day ticks are at 0:00 UT.

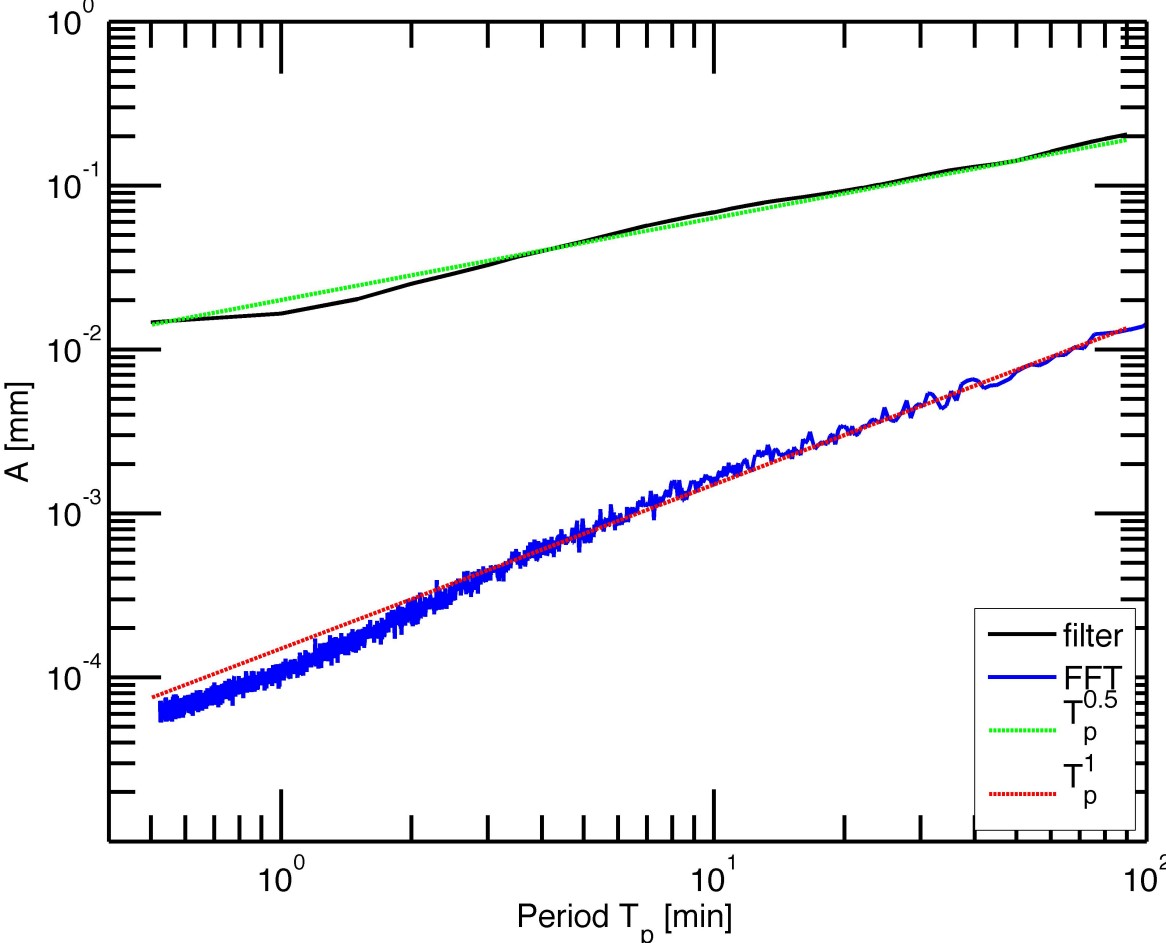

**Figure 2.** Amplitude spectra of IWV fluctuations observed at Bern in summer 2010. The solid black line is obtained by a band pass filtering method. The green line shows the inclination for $A \sim T_p^{0.5}$ where $T_p$ is the period. The corresponding power law is given by $P \sim f^{-1}$ where $f$ is the frequency. The blue line is obtained by a Fast Fourier Transform of the complete summer 2010 time interval. The red line shows the inclination for $A \sim T_p^{1}$ (or power $P \sim f^{-2}$). The FFT method underestimates the amplitudes and overestimates the steepness of the inclination.

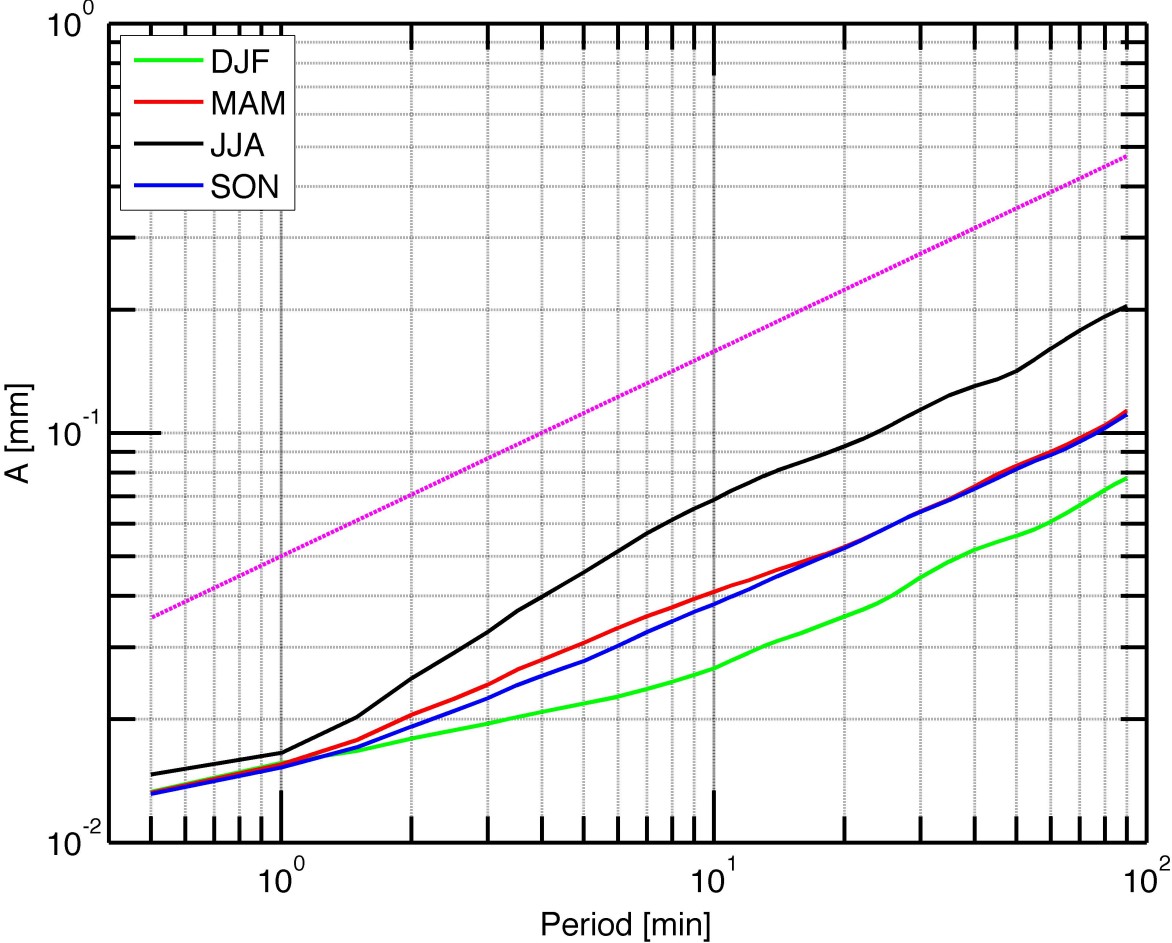

**Figure 3.** Amplitude spectra of IWV fluctuations observed at Bern for the four seasons of the year 2010. The green, red, blue and black line is obtained by a band pass filtering method for winter, spring, autumn and summer respectively. The magenta line shows the inclination for $A \sim T_p^{0.5}$ (or power $P \sim f^{-1}$) where $f$ is the frequency and $T_p$ is the period.

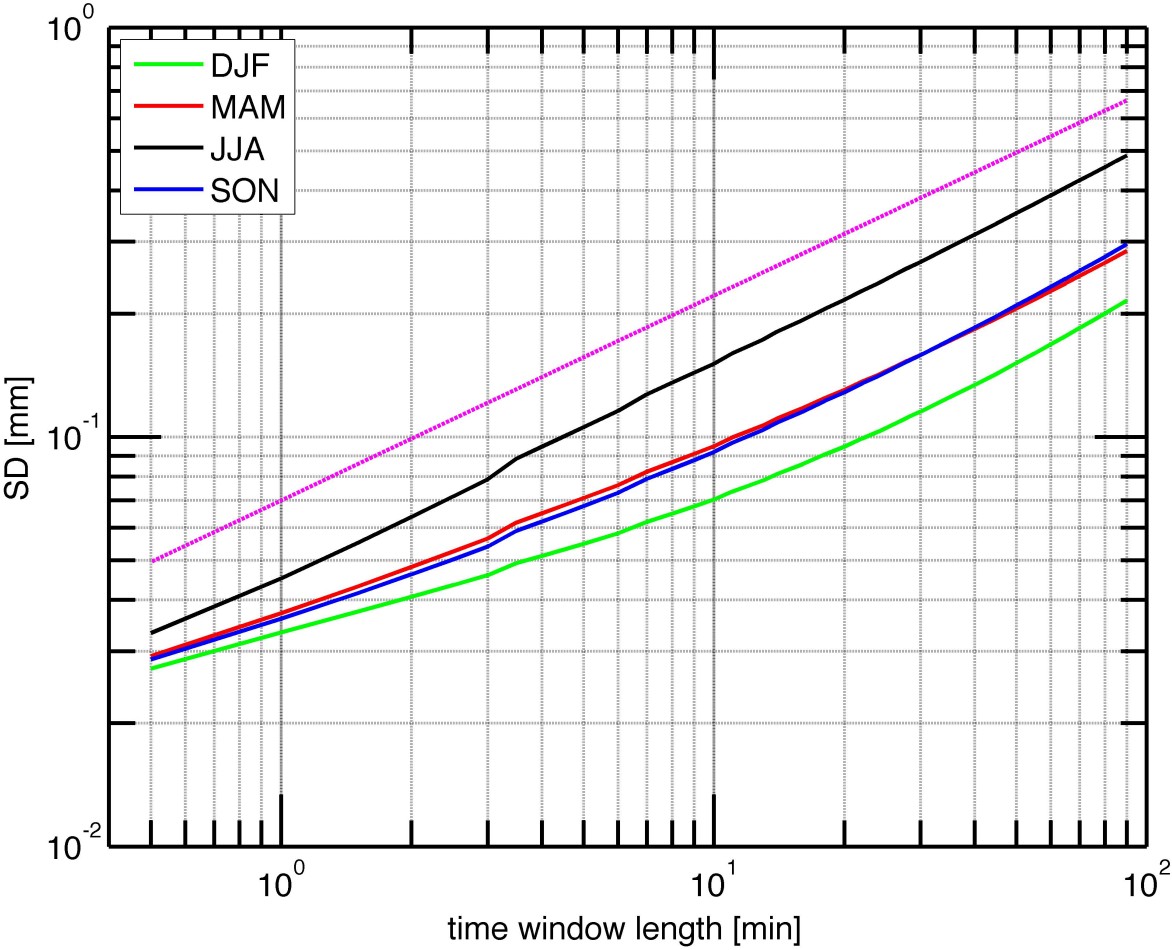

**Figure 4.** Spectra of IWV fluctuations observed at Bern for the four seasons of the year 2010. The green, red, black and blue line is obtained by a moving standard deviation (SD) with a variable time window length for winter, spring, summer and autumn respectively. The magenta line shows the inclination for $A \sim T_p^{0.5}$ (or power $P \sim f^{-1}$) where $f$ is the frequency and $T_p$ is the period.

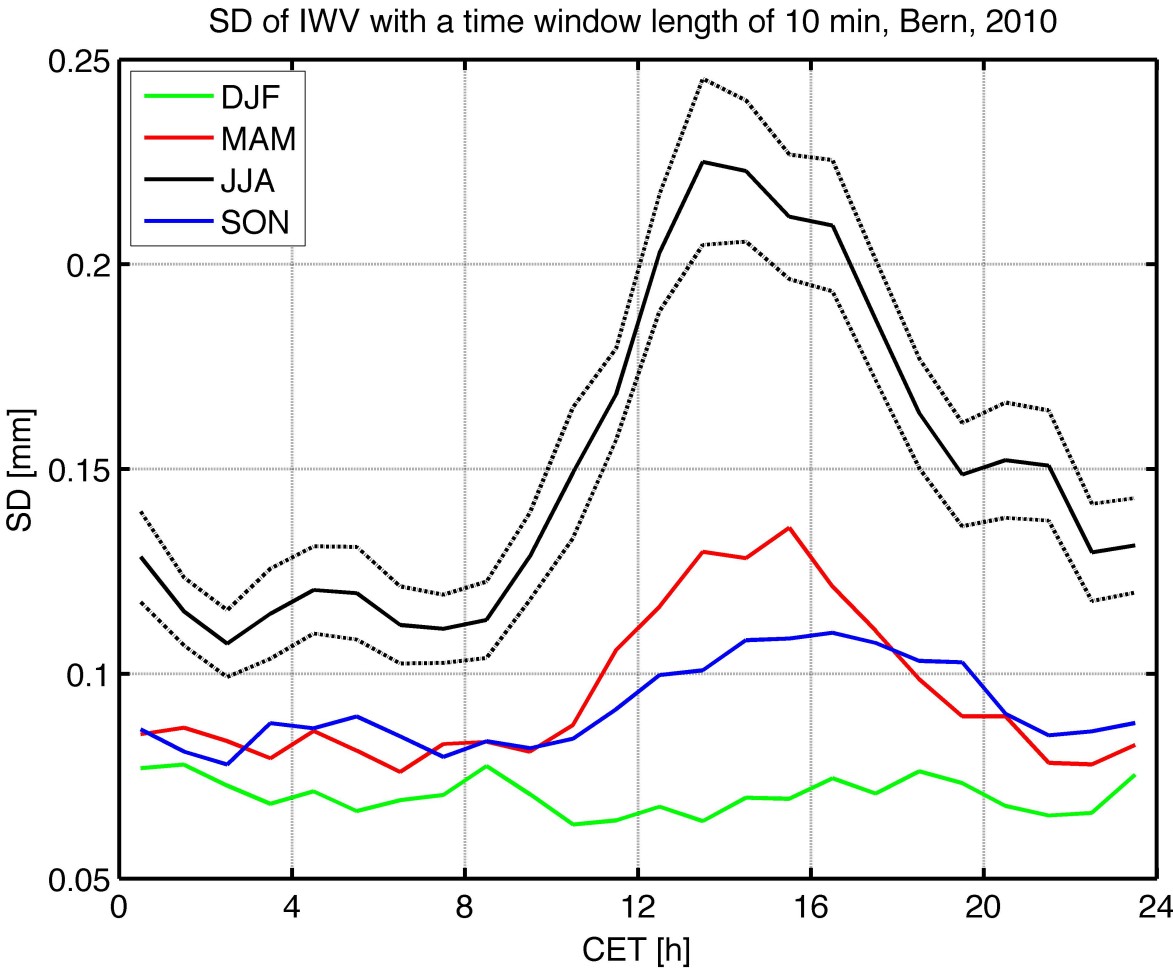

**Figure 5.** Diurnal cycle of IWV fluctuations observed at Bern for the four seasons of the year 2010 (moving standard deviation SD with time window length of 10 min). The green, red, blue and black line is for winter, spring, autumn and summer respectively. CET stands for Central European Time (Universal Time +1 hour). The error of the mean (for hourly averages) is shown for the summer season (JJA) by the dashed black lines.

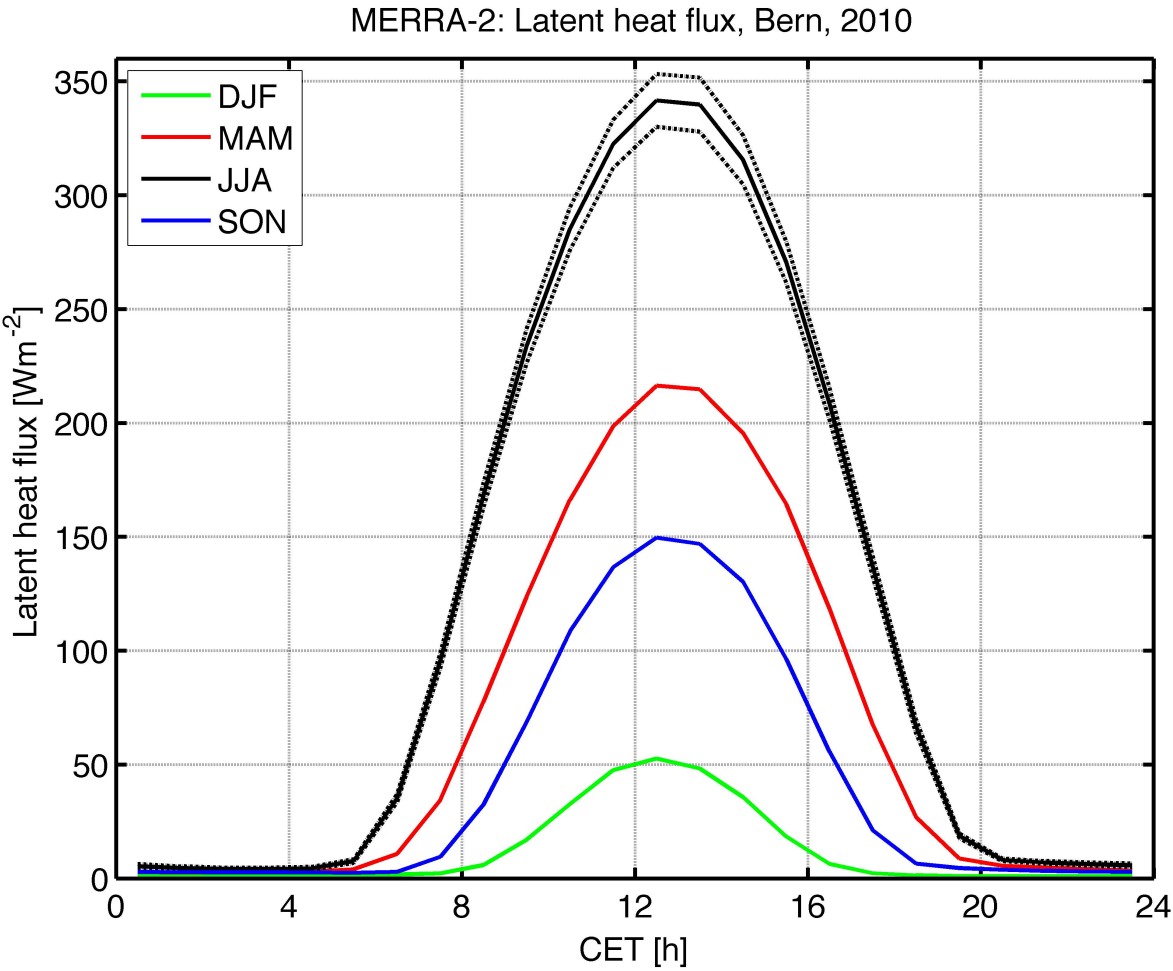

**Figure 6.** Diurnal cycle of latent heat flux near to Bern for different seasons as derived from MERRA-2 reanalysis data. The error of the mean (for hourly averages) is shown for the summer season (JJA) by the dashed black lines.

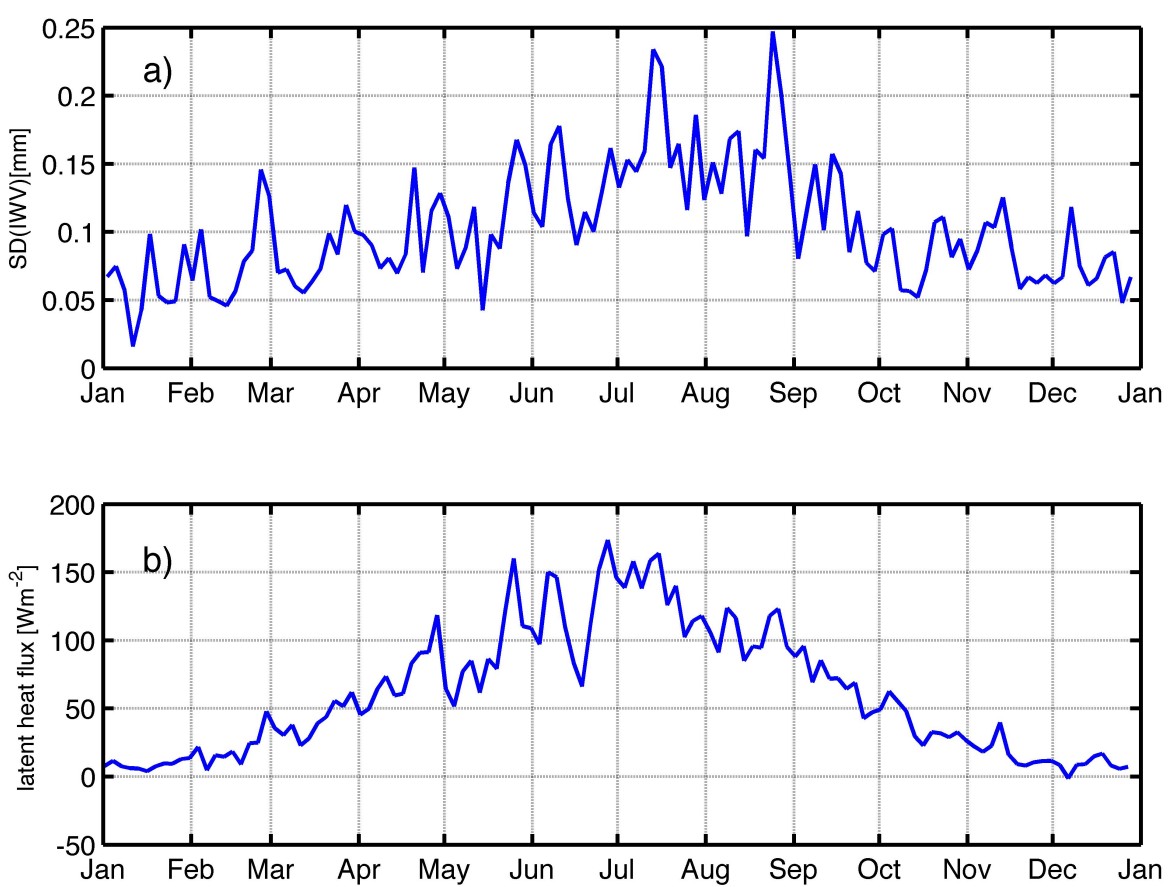

**Figure 7.** a) 3-day averages of the magnitude of the short-term IWV variability (moving SD of IWV with 10 min window) observed by TROWARA at Bern in 2010. 3-day averages of latent heat flux near to Bern derived from MERRA-2 reanalysis data in 2010.

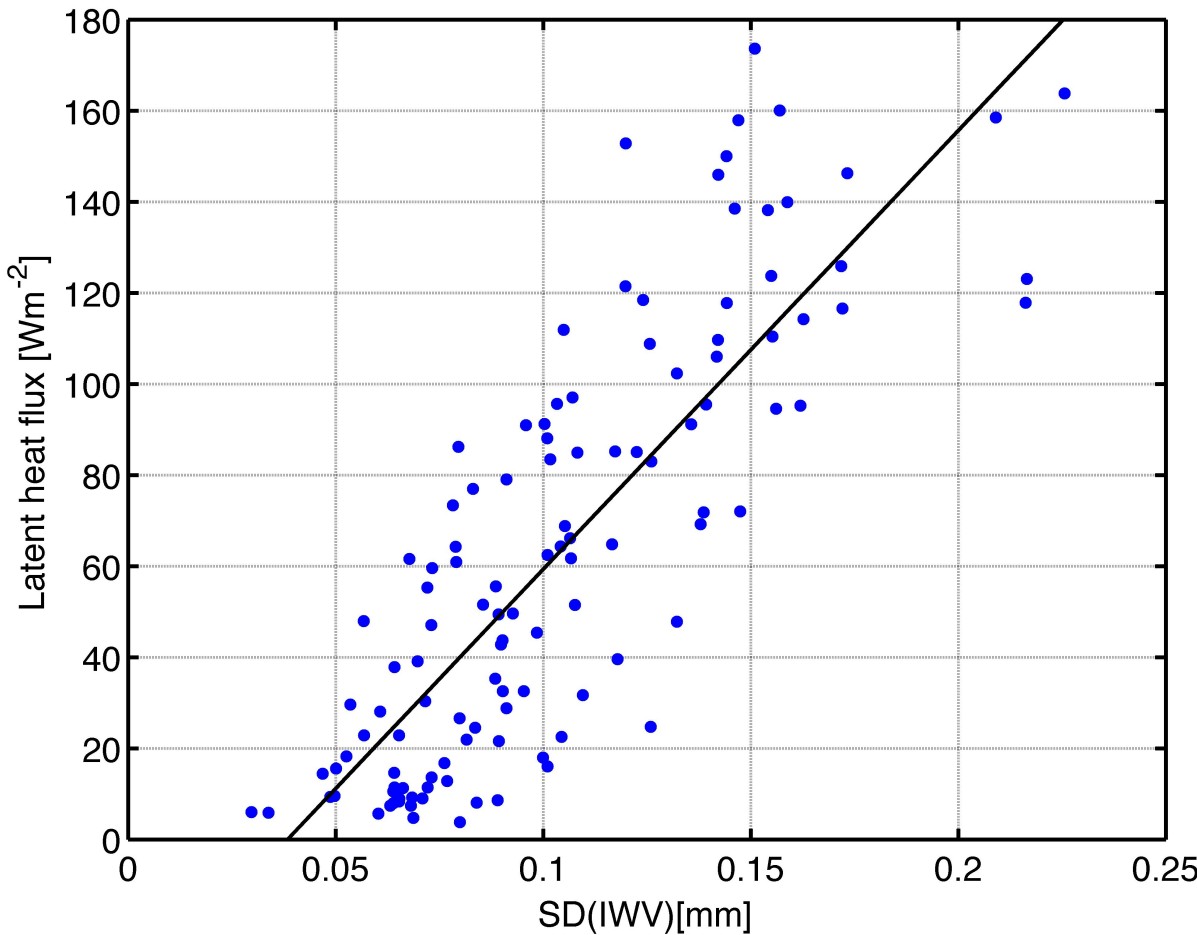

**Figure 8.** Scatter plot of SD(IWV) and the latent heat flux measured by TROWARA at Bern in 2010. The parameters are described in detail in Fig.7.