# Peer review of "Diurnal cycle of short-term fluctuations of integrated water vapour above Switzerland"

_Atmospheric Chemistry and Physics, 2019_

## Referee Comment (RC1) · Anonymous Referee #1 · 29 Apr 2019

**1   Content**

This manuscript describes short-term fluctuations of integrated water vapor (IWV) obtained by radiometric measurements at the University of Bern (Switzerland). The authors tested three different methods to derive the amplitude spectra of the IWV fluctuations and found that the method using the moving standard deviation give the best comparable results. Furthermore, the fluctuations are described and a changes in the fluctuations are attributed to the occurrence of turbulence.

[Figure]

**2  Overall impression and rating**

The overall impression of the manuscript is rather moderate. The presentation of the manuscript with all the figures is good in general and the text is easy to understand. However, the analysis is unfortunately not really done in a balanced way and I found most of the results to speculative without any further analysis and explanation (see my major comments). The paper has a strong focus on the methodology of evaluating fluctuations in general (4.5 pages out of 6), whereas the interpretation of the fluctuations which is suggested by the title is rather very short. With such a strong focus on the methodology the manuscript would better fit into the scope of AMT. Nevertheless, the data and their analysis itself are an important contribution to the community. For these reasons, I recommend publication in ACP after major revisions and an expansion of discussion part.

**3  Major comments:**

- First, I would like to mention the introduction. A complete motivation of the importance of water vapor and IWV with citation of the key papers is missing, e.g. Why is water vapor an import trace gas in the atmosphere (hydrological cycle etc.); Why is it important to understand IWV fluctuations ? Open questions ?. The existing literature concerning IWV fluctuations is poorly cited and the work described here is not really put into the context. For example missing papers are: Ortiz de Trenberth et al. 1998, Galisteo et al. 2011, 2014, Vogelmann et al. 2015 and others. I recommend to revise the introduction and add a general part about water vapor, IWV and its importance.

- Second, the manuscript is not very balanced in general. The method part including the introduction is about 4.5 pages, whereas the results section is only about

1 page describing 7 figures in total with only a sparse discussion and explanation. I found much of this text to be poorly supported. I would recommend to split the results section into a more method based sub-section concerning how to extract the best amplitude spectrum (FFT, band-pass, moving SD) and a sub-section concerning correlation of turbulence with fluctuations of IWV. This would help to better underlay both parts of the analysis (method and turbulence) with more explanation and more extensive analysis. As an example, in the text to figure 2-4 mainly the behavior of the spectra in comparison to a power law is described, but no discussion about which power spectra is expected and why do the spectra behave like we see it in the Figures.

- Third, there is a more detailed analysis missing to better understand the correlation of the IWV fluctuations and fluctuations of specific kinetic energy. It is obvious that the shape of both diurnal cycles is similar, but as you mentioned in the text the amplitude in different seasons is different. There are stronger fluctuations of specific kinetic energy in spring and you attributed this to stronger advection (page 6, line 15). This could be, but it is just speculation without any prove. From the diurnal cycle it is obvious that there are other processes involved influencing the IWV fluctuations. For example you could look into a connection between IWV and ILW fluctuations to determine the influence of clouds. This is also suggested by the cloud picture in Figure 1. Another example is that the IWV fluctuations in summer show also enhanced values during nighttime compared to other season, which is not reflected by the diurnal cylce of specific kinetic energy. In the end, I would like to have a more detailed discussion about possible other influencing factors like advection, cloud formation, precipitation, gravity waves etc.. All of this is not addressed at all in the manuscript.

**4  Specific comments/questions:**

- Page 3, line 32: You mentioned the rotational transition line of water vapour centered at 22.232 GHz. Why does this effect you measurements, because the microwave channel at 21.4GHz has only a bandwidth of 100 MHz.

- Page 5, line 1: Which length of the Hamming window do you use ? Maybe it is worth to mention this in the text.

- Page 6: Did you analysed the spectrum of the specific kinetic energy fluctuations ? The time resolution is of course lower than of the IWV fluctuations, but you could compare the slope of the spectra in comparison to the spectra of the IWV fluctuations for time window length larger than 10 min.

**5  Technical comments/suggestions:**

- page 1, line 7: calculation instead of computation

- page 1, line 14: "used" instead of regarded

- page 3, line 1: provide instead of give

- page 6, line 20: "explain" this is not shown at all, you should better replace the wording and use "indicate"

- Figure 1b: The caption of Figure 1b) could better describe the content of the Figure and should not contain the interpretation only.

**6  References:**

- Ortiz de Galisteo, J. P., Cachorro, V. , Toledano, C. , Torres, B. , Laulainen, N. , Bennouna, Y. and de Frutos, A. (2011), Diurnal cycle of precipitable water vapor over Spain. Q.J.R. Meteorol. Soc., 137: 948-958. doi:10.1002/qj.811

- Ortiz de Galisteo, J. P., Bennouna, Y. , Toledano, C. , Cachorro, V. , Romero, P. , Andrés, M. I. and Torres, B. (2014), Analysis of the annual cycle of the precipitable water vapour over Spain from 10‐year homogenized series of GPS data. Q.J.R. Meteorol. Soc., 140: 397-406. doi:10.1002/qj.2146

- Trenberth, K. E.: Atmospheric Moisture Residence Times and Cycling: Implications for Rainfall Rates and Climate Change, Clim. Change, 39, 667–694, doi:10.1023/A:1005319109110, 1998.

- Vogelmann, H., Sussmann, R., Trickl, T., and Reichert, A.: Spatiotemporal variability of water vapor investigated using lidar and FTIR vertical soundings above the Zugspitze, Atmos. Chem. Phys., 15, 3135-3148, https://doi.org/10.5194/acp-15-3135-2015, 2015.
* * *

---

## Referee Comment (RC2) · Anonymous Referee #3 · 17 Jul 2019

The paper describes the short term fluctuations in the integrated water vapour (IWV) column over Bern observed by the ground based TROWARA radiometer. Three different methods to calculate amplitude spectra of these fluctuations are presented. The main result of this work is an analysis of the diurnal cycle of short fluctuations and how this varies with season. During summer, the authors conclude through the similarities of this diurnal cycles to the one of the short term fluctuations of specific kinetic energy, that the observed short term fluctuations of the integrated water vapour column are caused by turbulence associated with convective heating.

The manuscript represent a substantial contribution to scientific progress within the scope of this journal because it provides a detailed analyses of the the seasonal and diurnal variations of the IWV fluctuations.

[Figure]

**Major points**

Most critical I see the conclusion, " that the diurnal cycle of the short-term IWV fluctuations is caused by turbulence associated with large convective heating during daytime in summer" from the comparison in this work, because:
- The diurnal cycles are not very similar (e.g. for JJA Figure 5 shows the maximum between 12 and 14h, Figure 6 after 16h.)
- I would expect a more detailed analysis of this connection, looking into more detail than the seasonal mean of the diurnal cycle. How is the correlation between them (for single days and/or do days with strong (week) short-term IWV fluctuations show also strong (weak) fluctuations of the specific kinetic energy)?
- What about spring and autumn?
- How large is the variability of the diurnal cycles shown in Figure 5 and 6?
I agree, that the convective heating could have a large influence on the diurnal cycle of the IWV short term fluctuations, but from the results presented I cannot see why it is the only/main cause?

**Other major points**

The results and discussion part is rather short and should be extended.

Why is the method of the moving standard deviation chosen to analyse the diurnal cycle, what is its advantage compared to the band pass filter?

What are the potential benefits for modelling studies from these measurements? In the introduction more literature concerning this topic should be mentioned.

**Minor points**

Page 2, line 33: "...height of the atmospheric boundary layer" instead of "...height level of the atmospheric boundary layer"?

Page 4, line 19: 11 seconds or 10 seconds as mentioned at page 2 line 1?

The link between the short term fluctuations of the specific kinetic energy an the turbulent kinetic energy could be explained in more detail.

Why does Figure 7 use the climatology and not the 2010 data as the rest of the paper?

---

## Author Comment (AC1) · 9 Aug 2019

Please see the pdf file in the supplement!

Please also note the supplement to this comment:
https://www.atmos-chem-phys-discuss.net/acp-2019-129/acp-2019-129-AC1-supplement.pdf
* * *

---

## Author Response (AR1)

Dear Reviewers and Editor,

We thank you for your helpful comments and corrections which were important to improve the article during a moderate or major revision.

(Reviewer comment is in blue,
our answer is in black,
manuscript change is in red)

**Reviewer 1:**
Point-to-point response:

1) However, the analysis is unfortunately not really done in a balanced way and I found most of the results to speculative without any further analysis and explanation (see my major comments). The paper has a strong focus on the methodology of evaluating fluctuations in general (4.5 pages out of 6), whereas the interpretation of the fluctuations which is suggested by the title is rather very short.

We changed our analysis: Instead of specific kinetic energy fluctuations, we take the diurnal cycle of latent heat flux at Bern as provided by MERRA-2 reanalysis. So, we analyse and discuss the relation of a cause (diurnal cycle of latent heat flux) to the short-term variability of IWV. Figure 6 and 7 are new and indicate that short-term variability of IWV is a consequence of the diurnal cycle in latent heat flux.

- 2) First, I would like to mention the introduction. A complete motivation of the im- portance of water vapor and IWV with citation of the key papers is missing, e.g. Why is water vapor an import trace gas in the atmosphere (hydrological cycle etc.); Why is it important to understand IWV fluctuations ? Open questions ?. The existing literature concerning IWV fluctuations is poorly cited and the work described here is not really put into the context. For example missing papers are: Ortiz de Trenberth et al. 1998, Galisteo et al. 2011, 2014, Vogelmann et al. 2015 and others. I recommend to revise the introduction and add a general part about water vapor, IWV and its importance.

We agree and added a paragraph about the role of atmospheric water vapour.

Atmospheric water vapour is the dominant greenhouse gas and acts like a warm mantle for the Earth. Global warming due to man-made $CO_2$ emissions is amplified by increase of the water vapour concentration in a warmer world. This amplification of global warming due to the so-called water vapour feedback is up to a factor of three \citep{held2000}. The electric dipole of the water molecule is responsible for the large latent heat of vaporization of water and for the strong interaction of electromagnetic waves with water vapour. Integrated water vapour (IWV) is the main contributor to the wet delay of signals of the Global Navigation Satellite System (GNSS) \citep{guerova2016}. Further, atmospheric water vapour is the reservoir gas for formation of cloud liquid water and precipitation such as snow, hail and rain which are relevant for weather and climate. The annual variation of integrated water vapour is rather strong at Bern (Switzerland) reaching from about 8 mm (or 8 kg m$^{-2}$) in

winter to 24 mm in summer \citep{hocke2017}. \cite{ortiz2014} reported that IWV ranges from 14.5 mm to 20.0 mm in Spain. Long-term monitoring of IWV is essential for detection of regional and global trends of IWV \citep{morland2009,parracho2018}.

- 3) Second, the manuscript is not very balanced in general. The method part includ- ing the introduction is about 4.5 pages, whereas the results section is only about 1 page describing 7 figures in total with only a sparse discussion and explanation. I found much of this text to be poorly supported. I would recommend to split the results section into a more method based sub-section concerning how to extract the best amplitude spectrum (FFT, band-pass, moving SD) and a sub-section concerning correlation of turbulence with fluctuations of IWV. This would help to better underlay both parts of the analysis (method and turbulence) with more ex- planation and more extensive analysis. As an example, in the text to figure 2-4 mainly the behavior of the spectra in comparison to a power law is described, but no discussion about which power spectra is expected and why do the spectra behave like we see it in the Figures.

We agree and we structure the result sections into three subsections. We added a few sentences in the result section. However, the relation of the diurnal cycle of latent heat flux and the diurnal cycle of IWV fluctuation strength is quite basic and we think that a long discussion is not needed. The slope of the IWV spectra was not yet simulated or observed. Thus we cannot discuss our expectation or compare it to previous results. Our observational study can be regarded as a letter which presents a new result, namely the diurnal cycle of short-term IWV fluctuations.

Figure \ref{fig6} shows the diurnal cycle of the latent heat flux in winter, spring, summer and fall as derived from MERRA-2 reanalysis data close to Bern in the year 2010. As expected, the diurnal cycle of latent heat flux is strongest in summer.
  The maximum latent heat flux is around 13:00 CET. That means the observed diurnal cycle of the short-term IWV fluctuations lags the diurnal cycle of latent heat flux by 1-2 hours. A numerical simulation by \cite{schlemmer2011} showed that the diurnal variation in solar short wave radiation leads the diurnal cycle of latent heat flux which is followed by the diurnal cycles of CAPE (convective available potential energy) and the convective mass flux. The maxima of the curves in CAPE and convective mass flux occurred in the afternoon and evening hours.

  These observations and simulations suggest that the increase of short-term IWV fluctuations during daytime in summer is due to the diurnal cycle of latent heat flux which is a precondition for an increase of strong and variable convection cells in the afternoon. The up- and downdraft regions of the convection cells are passing the antenna beam of the TROWARA radiometer which consequently measures larger or smaller values of IWV in time distances of about 10 min. In addition, a convection cell is itself time variable: The lifetime of a convection cell is roughly between 30 and 60 minutes \citep{giaiotti2007}.

 Figure \ref{fig7} compares the 3-day averages of SD of IWV (moving 10 min-window) in the upper panel with 3-day averages of MERRA-2 latent heat flux at Bern in the lower panel for the year 2010. The correlation coefficient $r$ of SD(IWV) and the latent heat flux is equal to 0.82. Some of the peaks in the latent heat flux coincidently appear in the SD curve of

short-term IWV variability in the upper panel, e.g., the double peak around June 2010. Figure \ref{fig7} suggests that the variability of the latent heat flux is a contributor to the short-term variability of IWV.

4) Third, there is a more detailed analysis missing to better understand the correla- tion of the IWV fluctuations and fluctuations of specific kinetic energy. It is obvious that the shape of both diurnal cycles is similar, but as you mentioned in the text the amplitude in different seasons is different. There are stronger fluctuations of specific kinetic energy in spring and you attributed this to stronger advection (page 6, line 15). This could be, but it is just speculation without any prove. From the diurnal cycle it is obvious that there are other processes involved influencing the IWV fluctuations. For example you could look into a connection between IWV and ILW fluctuations to determine the influence of clouds. This is also suggested by the cloud picture in Figure 1. Another example is that the IWV fluctuations in summer show also enhanced values during nighttime compared to other season, which is not reflected by the diurnal cylce of specific kinetic energy. In the end, I would like to have a more detailed discussion about possible other influencing factors like advection, cloud formation, precipitation, gravity waves etc.. All of this is not addressed at all in the manuscript.

We agree, the comparison of the diurnal cycle of short-term IWV fluctuations with specific kinetic energy (TKE) fluctuations was not so reasonable since TKE also includes the wind interaction with the surface which can be strong during equinox. In addition, small-scale turbulence may not induce significant changes in integrated water vapour. Thus, TKE fluctuations and IWV fluctuations are possibly not so much related. We argue now in a stronger manner that IWV fluctuations mainly depend on the spatio-temporal variability of convection cells and the amount of integrated water vapour in the updraft and downdraft regions. However, we added a sentence that other contributors to the IWV variability exist. One can see in the new Figure 7, that some peaks of SD(IWV) are not included in the annual curve of the latent heat flux.
The relation of the diurnal cycles of varios parameters are well discussed in the simulation study of Schlemmer et al. (2001). The diurnal cycle of latent heat flux in the reanalysis data agrees with this study. The phase lag of the diurnal cycle of SD(IWV) appears reasonable since the diurnal cycle in cloud liquid water (ILW) is also phase lagged with respect to the diurnal cycle of latent heat flux. Usually, ILW fluctuations have no influence on IWV but both, ILW fluctuations and IWV fluctuations, are connected to atmospheric convection.

Figure 6 and 7 are new. The discussion is now easier.

• Page 3, line 32: You mentioned the rotational transition line of water vapour cen- tered at 22.232 GHz. Why does this effect you measurements, because the microwave channel at 21.4GHz has only a bandwidth of 100 MHz.

The pressure broadening of the 22 GHz line also influences the intensity at 21 GHz which is in the line wing

• Page 5, line 1: Which length of the Hamming window do you use ? Maybe it is worth to mention this in the text.

We added a sentence that the size of the Hamming window is equal to the number of filter coefficients (three times of central period).

- Page 6: Did you analysed the spectrum of the specific kinetic energy fluctuations ? The time resolution is of course lower than of the IWV fluctuations, but you could compare the slope of the spectra in comparison to the spectra of the IWV fluctuations for time window length larger than 10 min.

The new manuscript version do not consider the specific kinetic energy fluctuations.

Thank you for the Technical components which we included in the new version!

**Reviewer 3:**
Point-to-point response:

1) Most critical I see the conclusion, " that the diurnal cycle of the short-term IWV fluctua- tions is caused by turbulence associated with large convective heating during daytime in summer" from the comparison in this work, because:
- The diurnal cycles are not very similar (e.g. for JJA Figure 5 shows the maximum between 12 and 14h, Figure 6 after 16h.)

We agree. When we submitted the paper we were also not happy about the weak correlation between turbulence (TKE fluctuations) and IWV fluctuation strength.
After your comments, we got the idea that the diurnal cycle of latent heat flux as provided by MERRA-2 reanalysis is the cause of the observed diurnal cycle of SD(IWV). Both curves must be not in phase. The study of Schlemmer et al. (2001) shows for example that the maximum of latent heat flux occurs around noon while the cloud liquid water and precipitation peaks in the afternoon and early evening hours. This is possibly due to the ascent time of water vapour.

Fig. 6 and 7 are new. We found a strong correlation of the annual curves of latent heat flux and SD(IWV). We changed the discussion.

I would expect a more detailed analysis of this connection, looking into more detail than the seasonal mean of the diurnal cycle. How is the correlation between them (for single days and/or do days with strong (week) short-term IWV fluctuations show also strong (weak) fluctuations of the specific kinetic energy)?
- What about spring and autumn?
- How large is the variability of the diurnal cycles shown in Figure 5 and 6?

Thank you for your fruitful comments which we considered in the revision.

Figure 7 gives an overview on the variability of 3-day averages of latent heat flux and SD(IWV). We emphasize that single peaks in latent heat flux coincidently appear in the observed SD(IWV).

We show the error of the mean in Figures 5 and 6 for the season JJA.

Figure \ref{fig6} shows the diurnal cycle of the latent heat flux in winter, spring, summer and fall as derived from MERRA-2 reanalysis data close to Bern in the year 2010. As expected, the diurnal cycle of latent heat flux is strongest in summer.
  The maximum latent heat flux is around 13:00 CET. That means the observed diurnal cycle of the short-term IWV fluctuations lags the diurnal cycle of latent heat flux by 1-2 hours. A numerical simulation by \cite{schlemmer2011} showed that the diurnal variation in solar short wave radiation leads the diurnal cycle of latent heat flux which is followed by the diurnal cycles of CAPE (convective available potential energy) and the convective mass flux. The maxima of the curves in CAPE and convective mass flux occurred in the afternoon and evening hours.

  These observations and simulations suggest that the increase of short-term IWV fluctuations during daytime in summer is due to the diurnal cycle of latent heat flux which is a precondition for an increase of strong and variable convection cells in the afternoon. The up- and downdraft regions of the convection cells are passing the antenna beam of the TROWARA radiometer which consequently measures larger or smaller values of IWV in time distances of about 10 min. In addition, a convection cell is itself time variable: The lifetime of a convection cell is roughly between 30 and 60 minutes \citep{giaiotti2007}.

  Figure \ref{fig7} compares the 3-day averages of SD of IWV (moving 10 min-window) in the upper panel with 3-day averages of MERRA-2 latent heat flux at Bern in the lower panel for the year 2010. The correlation coefficient $r$ of SD(IWV) and the latent heat flux is equal to 0.82. Some of the peaks in the latent heat flux coincidently appear in the SD curve of short-term IWV variability in the upper panel, e.g., the double peak around June 2010. Figure \ref{fig7} suggests that the variability of the latent heat flux is a contributor to the short-term variability of IWV.

The results and discussion part is rather short and should be extended.

We added several sentences. The discussion is now easier since latent heat flux and SD(IWV) are related. So, the discussion can be kept short. We refer to the study of Schlemmer et al. (2001). Finally, we like that the manuscript is more like a letter.

Why is the method of the moving standard deviation chosen to analyse the diurnal cycle, what is its advantage compared to the band pass filter?

We added a sentence:
In the following, we focus on the third method (SD values) since the standard deviation is most common for the characterization of the variability of a time series.

What are the potential benefits for modelling studies from these measurements? In the introduction more literature concerning this topic should be mentioned.

A potential benefit is that high resolution atmosphere modelling may produce a similar diurnal cycle of short-term IWV fluctuation strengths as we observed. This would be a confirmation that atmospheric convective processes are well represented in the model world. We added the following paragraph at the end of the manuscript:

Thus, we suggest that the diurnal cycle of the short-term IWV fluctuations is mainly caused by the diurnal cycle of latent heat flux which is a precondition for strong and variable convection cells in the afternoon during summer. The spatio-temporal variability of the convection cells induces the diurnal cycle of short-term IWV fluctuations as observed by the TROWARA radiometer at Bern in summer. However, other sources such as eddies in the lower troposphere may also contribute to the short-term variablility of IWV. High resolution modelling of the diurnal cycle of short-term IWV fluctuations could be compared to the TROWARA observations so that one can estimate how good convective processes are represented by the model. Generally, we think that the high resolution IWV observations of TROWARA can contribute to research on the atmospheric boundary layer.

Page 2, line 33: "...height of the atmospheric boundary layer" instead of "...height level of the atmospheric boundary layer"?

We agree and we changed it

Page 4, line 19: 11 seconds or 10 seconds as mentioned at page 2 line 1?

Now, we keep 10 seconds.

The link between the short term fluctuations of the specific kinetic energy an the turbulent kinetic energy could be explained in more detail.

we do not need this link in the new version.

Why does Figure 7 use the climatology and not the 2010 data as the rest of the paper?

Figure 7 is replaced by a different figure.

Thank you for your review!

[revised manuscript text omitted]

---

## Referee Report (RR1)

**Review of the Revision: Diurnal cycle of short-term fluctuations of integrated water vapour above Switzerland by Hocke et al.**

**Overall impression and rating**

The revised manuscript improved in a considerably way and gives overall a better impression. The introduction improved and the restructuring of the results section makes it easier to read. The analysis is still quite short, but sufficient and written in more balanced way. For these reasons, I recommend publication in ACP after considering my minor comments.

**Minor comments:**

- As a motivation of using MERRA2 for latent heat flux, I would recommend to cite the DRAPER et al. 2018 study. They did a lot of work concerning representation of latent heat in MERRA2.

- The diurnal cycle of latent heat flux in winter time shows also a maximum around noon time, which is of course much weaker (~ 7 times) compared to summer. This enhancement is not visible at all in the IWV fluctuation. Can you please explain why this is the case and please discuss that also in the manuscript.

- I appreciate that you included statistical information about the IWV fluctuations and latent heat flux correlation in terms of the correlation coefficient. Could you please add also information about the significance (e.g. p-value) of the correlation. Please show also a correlation scatter plot with IWV fluctuations on the x-axis and latent heat flux on the y-axis. The plot allows the reader to better see how the correlation looks like and should further support the main massage of the paper.

**Technical comments/suggestions:**

- page 2, line 30: Citation is missing

- page 7, line 14: drives instead of leads ?

**References:**

- Draper, C.S., R.H. Reichle, and R.D. Koster, 2018: Assessment of MERRA-2 Land Surface Energy Flux Estimates. J. Climate, 31, 671–691, https://doi.org/10.1175/JCLI-D-17-0121.1

---

## Author Response (AR2)

**Point-to-point response to referee 1, second revision:**
(our response is in blue, the changed text is in red)

**Overall impression and rating**

The revised manuscript improved in a considerably way and gives overall a better im- pression. The introduction improved and the restructuring of the results section makes it easier to read. The analysis is still quite short, but sufficient and written in more balanced way. For these reasons, I recommend publication in ACP after considering my minor comments.

Thank you!

**Minor comments:**

As a motivation of using MERRA2 for latent heat flux, I would recommend to cite the DRAPER et al. 2018 study. They did a lot of work concerning represen- tation of latent heat in MERRA2.

This is a good idea. We added the reference and a sentence:

\cite{draper2018} investigated the surface energy fluxes in MERRA-2 in detail and obtained a positive bias of about 5 Wm$^{-2}$ for the latent heat flux of global land annual averages.

The diurnal cycle ofl atentheatfluxinwintertimeshowsalsoamaximumaround noon time, which is of course much weaker ($\sim$ 7 times) compared to summer. This enhancement is not visible at all in the IWV fluctuation. Can you please explain why this is the case and please discuss that also in the manuscript.

Yes, we add a short paragraph which mentions the open question why there is no maxima of IWV fluctuations around noon in winter.

It remains an open question why the IWV fluctuations do not show a maximum around noon in winter in Fig.\ref{fig5 } though there is a clear maximum in latent heat flux in winter around noon (Fig. \ref{fig6}). It could be that other processes such as friction of surface winds and turbulence may play a role in winter for generation of IWV fluctuations so that latent heat flux is not the dominant factor in winter.

I appreciate that you included statistical information about the IWV fluctuations and latent heat flux correlation in terms of the correlation coefficient. Could you please add also information about the significance (e.g. p-value) of the corre- lation.

We added a entence about the 95% confidence interval of the correlation.

Further, the curves of SD(IWV) and the latent heat flux yield a correlation coefficient $r$ of 0.82 at Bern in 2010 while the 95\% confidence interval ranges from 0.75 to 0.87.

Please show also a correlation scatter plot with IWV fluctuations on the x-axis and latent heat flux on the y-axis. The plot allows the reader to better see how the correlation looks like and should further support the main message of the paper.

Good idea. We added the Figure 8 in the new version

Figure \ref{fig8} provides a scatter plot of SD(IWV) and the latent heat flux. The linear regression line is given by the black solid line. It is obvious that a linear dependence exists between both parameters. Because of instrumental noise in the SD(IWV) values the linear regression line intersects the y-axis at -37.0 Wm$^{-2}$.

**Technical comments/suggestions:**

• page 2, line 30: Citation is missing

Schlemmer et al. (2011) is added

• page 7, line 14: drives instead of leads ?

yes, we use the word "drives"

[revised manuscript text omitted]